# Population Density and Host Preference of the Japanese Pine Sawyer (*Monochamus alternatus*) in the Qinling–Daba Mountains of China

**DOI:** 10.3390/insects14020181

**Published:** 2023-02-13

**Authors:** Junke Nan, Jingyu Qi, Yuexiang Yang, Mengqin Zhao, Chaoqiong Liang, Hong He, Cong Wei

**Affiliations:** 1Key Laboratory of Plant Protection Resources and Pest Management of the Ministry of Education, College of Plant Protection, Northwest A&F University, Yangling 712100, China; 2Key Laboratory of National Forestry and Grassland Administration for Control of Forest Biological Disasters in Western China, College of Forestry, Northwest A&F University, Yangling 712100, China; 3Shaanxi Academy of Forestry, Xi’an 710082, China

**Keywords:** *Monochamus alternatus*, pine wilt disease, population density, developmental stages, feeding preference, oviposition selection

## Abstract

**Simple Summary:**

*Monochamus alternatus* is not only a serious trunk-boring pest but also the most important and effective vector of the pine wood nematode. The effective control of *M. alternatus* can effectively limit the spread of pine wilt disease in the Qinling–Daba Mountains. However, little is known about the damage level and host preference of *M. alternatus* on different host plants. Here, we investigated the population density of *M. alternatus* overwintering larvae and revealed the host preference and oviposition selection of *M. alternatus* adults on *Pinus tabuliformis*, *P. armandii*, and *P. massoniana*. The results improve our understanding of the host preference of *M. alternatus* for feeding and oviposition.

**Abstract:**

*Monochamus alternatus* is a serious trunk-boring pest and is the most important and effective vector of the pine wood nematode *Bursaphelenchus xylophilus*, which causes pine wilt disease. The pine wilt disease poses a serious threat to forest vegetation and ecological security in the Qinling–Daba Mountains and their surrounding areas. In order to clarify whether the population density of *M. alternatus* larvae is related to the host preference of *M. alternatus* adults, we investigated the population density of *M. alternatus* overwintering larvae and explored the host preference of *M. alternatus* adults on *Pinus tabuliformis*, *P. armandii*, and *P. massoniana*. The results show that the population density of *M. alternatus* larvae was significantly higher on *P. armandii* than those on *P. massoniana* and *P. tabuliformis*. The development of *M. alternatus* larvae was continuous according to the measurements of the head capsule width and the pronotum width. Adults of *M. alternatus* preferred to oviposit on *P. armandii* rather than on *P. massoniana* and *P. tabuliformis*. Our results indicate that the difference in the population density of *M. alternatus* larvae between different host plants was due to the oviposition preference of *M. alternatus* adults. In addition, the instars of *M. alternatus* larvae could not be accurately determined, because Dyar’s law is not suitable for continuously developing individuals. This study could provide theoretical basis for the comprehensive prevention and control of the pine wilt disease in this region and adjacent areas.

## 1. Introduction

The Japanese pine sawyer, *Monochamus alternatus* Hope (Coleoptera: Cerambycidae), is mainly distributed in China, Japan, South Korea, North Korea, and several Southeast Asian countries [1,2]. It is a serious trunk-boring pest and is the most important and effective vector of the pine wood nematode *Bursaphelenchus xylophilus* (Steiner & Buhrer) Nickle [3]. The pine wilt disease caused by the pine wood nematode is a devastating forest disease that seriously endangers forestry safety across the world [4]. It can quickly kill thousands of pine trees in its invasive range, not only causing serious economic and ecological damage but also having a negative impact on all living things within the forest ecosystem [5]. The pine wood nematode is a major alien invasive species in China and has been listed among the quarantine objects of forest plants in China and many other countries [6,7]. The pine wood nematode is native to North America, and it has also spread to many other countries such as Japan, South Korea, Portugal, and Spain [7]. In China, the first case of *Pinus thunbergii* showing the pine wilt disease was found in the forests of Zijin Mountain of Nanjing, Jiangsu Province, in 1982 [8]. As of March 2022, the pine wilt disease had spread to 731 county-level administrative regions of 19 provinces, cities, or autonomous regions (Announcement No. 6, 2022, National Forestry and Grassland Administration).

The Qinling–Daba Mountains are an important geographical and ecological boundary in Central China [9], and they are also the watershed between the Yangtze River and the Yellow River. This area is an important hub of China’s geographical pattern, a large-scale east–west ecological corridor, and the area with the highest level of ecological security, which breeds rich natural animal and plant resources with a complex, diverse, and unique natural environment [10,11]. In the Qinling Mountains, the main susceptible pine species are *P. tabuliformis*, *P. armandii*, and *P. massoniana*. In the Daba Mountains, the main susceptible pine species is *P. massoniana*. In addition, *P. tabuliformis*, *P. armandii* and *P. massoniana* are widely distributed and have large stand areas in the Qinling–Daba Mountains, where the pine wilt disease was first discovered in Zhashui County of Shaanxi Province in 2009. As of March 2022, the pine wilt disease had spread to 171 township-level administrative regions of 24 county-level administrative regions in Shaanxi Province, and it currently covers an area of 321.4 km^2^, resulting in the death of more than 4 million pine trees (Announcement No. 5, 2022, Shaanxi Forestry Administration). The pine wilt disease poses a serious threat to forest vegetation and ecological security in the Qinling–Daba Mountains and surrounding areas.

It is necessary to better understand the influences of different hosts on the dispersal behavior of *M. alternatus* [12], which may provide information to effectively limit the spread of the pine wilt disease inrelated arreas. Many studies showed that the damage level of *M. alternatus* is not consistent due to different host plants. Tu et al. (2019) studied the effects of three host plants, *P. massoniana*, *P. elliottii* and *Cedarus cedarosa*, on the longevity and reproduction of *M. alternatus* adults, and found that the mass indoor breeding of *M. alternatus* adults on *P. massoniana* was beneficial to the reproduction of offspring compared to other two host plants [13]. Bonifácio et al. (2015) found that the selection of host plants by *M. alternatus* can be divided into two stages: one involves the newly emerged adults feeding on healthy trees, and the other involves the female adults preparing to lay eggs on injured or dead trees [14]. In either case, the selection of the host by *M. alternatus* is derived from the volatile chemical stimulation of the host pine trees [15]. Recently, Nan et al. (2021) found that the spatial distribution of *M. alternatus* larvae on different host plants differed in the Qinling–Daba Mountains [16]. However, studies of *M. alternatus* in the Qinling–Daba Mountains are extremely limited, which seriously hinders the prevention and control of the pine wilt disease in this region.

Here, the population density and the developmental stages of *M. alternatus* larvae were compared on different host plants (*P. tabuliformis*, *P. armandii*, and *P. massoniana*) killed by the pine wood nematode in the Qinling–Daba Mountains. In addition, the feeding preference and oviposition selection of *M. alternatus* adults on different host plants were investigated, which may provide useful information for the comprehensive prevention and control of the pine wilt disease in Qinling–Daba Mountains and their adjacent areas.

## 2. Materials and Methods

### 2.1. Experimental Field

According to the distribution of the pine wilt disease in the Qinling–Daba Mountains, Fenghuang Town, Yingpan Town and Xialiang Town of Zhashui County, Daheba Town of Foping County, and Chengguan Town of Ningshan County, located on the southern slope of the Qinling Mountains, and Chengguan Town and Zuolong Town of Langao County, as well as Yankou Town of Xixiang County, located on the northern slope of the Daba Mountains, were selected as the study areas (Table 1). In the experimental field, the main susceptible tree species are *P. tabuliformis*, *P. armandii* and *P. massoniana*, and the distance between the individually sampled trees on the same experimental site was relatively close (0–50 m).

### 2.2. Population Density and Development Stages of M. alternatus on Different Host Plants

After the specimens were collected, *M. alternatus* larvae and imago were observed under a stereomicroscope (OLYMPUS SZX16) and identified according to the morphology outlined by of Jiang (1989) [17] and Yang and Lin (2017) [18], respectively. In addition, genomic DNA was extracted from some young larvae which were difficult to distinguish morphologically from related species, and the mitochondrial *COI* gene was amplified and sequenced for molecular identification to ensure the identity was correct.

According to the distribution of dead pine trees caused by the pine wilt disease in the study area, a total of 38 dead pine trees of *P. tabuliformis* (10), *P. armandii* (14) and *P. massoniana* (14) (trees infested with the pine wood nematode, with a diameter at breast height of 16.82 ± 0.76 cm) were sampled at each study site in December of 2020. The dead trees killed by the pine wilt disease were felled from the ground level with a chainsaw and cut into 30–40 cm-long logs from the base of the trunk to the top of the tree (usually above 12 m), and they were numbered in sequence (Figure 1A). Then, we stripped the bark from the logs and split the logs into small splints with a diameter of no more than 2 cm, aiming to collect all the longicorn larvae. The longicorn larvae collected from all sections of the logs, including the bark (under which the young larvae lived), were recorded.

All *M. alternatus* larvae were brought back to the laboratory and identified through morphological and molecular identification. Two morphological traits, the head capsule width and the pronotum width, were used to estimate the developmental stage of *M. alternatus* larvae on the three host plants (Figure 1B) [19,20,21].

In addition, five other dead trees of each study site were felled from the ground level with a chainsaw and cut into 1 m-long logs from the base of the trunk to the top of the tree and placed into cages in the field (1.3 m × 0.9 m × 0.6 m), respectively. In the next year, the newly emerged *M. alternatus* adults were collected from these logs and recorded every week from the beginning to the end of the study.

### 2.3. Feeding and Oviposition Selection of M. alternatus Adults

In order to verify whether the population density and developmental stages of *M. alternatus* overwintering larvae on the three host pine species are related to their feeding preference and oviposition, we studied the larvae in Chengguan Town, Ningshan County, from July to September of 2021. In early July, three pairs of newly emerged males and females representing *M. alternatus* adults were placed into a cage (1.5 m × 1.0 m × 1.0 m), in which we placed fresh pine branches (length: 92.2 ± 1.6 cm) that consisted of young branches (one year old) and old branches (2–3 years old) inserted into bottles filled with water, as well as logs (length: 71.3 ± 0.7 cm, diameter: 26.3 ± 3.1 cm), of the three pine species (Figure 1C). The feeding behavior and oviposition of *M. alternatus* adults were regularly observed from July to September. The fresh pine branches and logs and *M. alternatus* adults were replaced every 3 days for a total number of 23 repetitions during the peak of the emergence of *M. alternatus* adults. The feeding part, the feeding area and the number of oviposition points on the logs of the three pine species were measured and recorded [22,23,24].
Feeding area (mm^2^) = length of feeding wound (mm) × width of feeding wound (mm);
Total feeding area (mm^2^) = total feeding area of young branches (one year old) (mm^2^) + total feeding area of old branches (2–3 years old) (mm^2^) + total feeding area of bark of logs (mm^2^); 
Feeding ratio of feeding part (%) = total feeding area of one feeding part (mm^2^)/total feeding area of one host pine tree (mm^2^) × 100%; 
Feeding ratio of host pine tree (%) = total feeding area of one host pine tree (mm^2^)/total feeding area (mm^2^) × 100%; 
The number of oviposition = the number of oviposition scars.

### 2.4. Statistical Analysis

All datasets were tested for normality and subjected to analysis of variance (ANOVA) using IBM SPSS statistics 22.0 software. Regression analysis was used to compare the correlations between data.

## 3. Results

### 3.1. Population Density and Developmental Stages of M. alternatus Overwintering Larvae on Different Host Plants

In total, 5304 overwintering larvae of *M. alternatus* were collected from 38 dead pine trees of *P. tabuliformis* (10), *P. armandii* (14), and *P. massoniana* (14) in December of 2020. There were averages of 235.50 ± 36.06 larvae on one *P. armandii*, 91.21 ± 18.80 larvae on one *P. massoniana*, and 73.00 ± 21.81 larvae on one *P. tabuliformis*, respectively (Table 2).

In addition, the size of the *M. alternatus* overwintering larvae on the three pine tree species also showed obvious differences. The head capsule width and pronotum width were measured to determine the developmental stages of the overwintering larvae, compared between different pine trees. The regression analyses indicated that there was a significant correlation between the head capsule width and pronotum width of larvae on the three host plants (*p* < 0.01), which showed a certain linear regression relationship. The regression equation for the head capsule width and pronotum width of the larvae was obtained as follows: *y* = 1.5929 *x* + 0.1020, *R*^2^ = 0.9209 (Figure 2B). However, the development of *M. alternatus* larvae was continuous based on their morphology and the measurements of the head capsule width and the pronotum width (Figure 2), indicating that *M. alternatus* larvae instars could not be accurately determined by the head capsule width and the pronotum width, because Dyar’s law is not suitable for continuously developing individuals. However, the developmental stages of *M. alternatus* larvae could be artificially divided into the five groups according to the head capsule width and the pronotum width (Appendix A). The ranges of the five groups are as follows: Head capsule width: Group 1: 0.51–1.40 mm, Group 2: 1.41–2.30 mm, Group 3: 2.31–3.20 mm, Group 4: 3.21–4.10 mm, Group 5: 4.11–5.00 mm; Pronotum width: Group 1: 1.11–2.50 mm; Group 2: 2.51–3.90 mm; Group 3: 3.91–5.30 mm; Group 4: 5.31–6.70 mm; and Group 5: 6.71–8.10 mm (Appendix A). The developmental stages of *M. alternatus* larvae showed obvious differences across the three pine species. Individuals of groups 4–5 were dominant on *P. armandii* (Figure 3E,F), individuals of groups 3–4 were dominant on *P. massoniana* (Figure 3C,D), and individuals of groups 2–3 were dominant on *P. tabuliformis* (Figure 3A,B). The size of *M. alternatus* larvae was observed to be the largest on *P. armandii*, followed by *P. massoniana* and *P. tabuliformis* (Figure 3).

### 3.2. The Emergence of M. alternatus Adults on Different Pine Trees

In the field observation of the logs, which began in the early May of 2021, the emergence of *M. alternatus* adults was first observed on *P. armandii* on 25 May (Figure 4), while the emergence time of the adults on *P. massoniana* and *P. tabuliformis* was one week later. In addition, the emergence number of *M. alternatus* adults on *P. armandii* (102) was far more than those of adults on *P. massoniana* (42) and *P. tabuliformis* (11) during the whole observation (Figure 4). The emergence time and the number of *M. alternatus* adults should be closely related to the population density and developmental stages of overwintering larvae on the pine trees.

### 3.3. Feeding Preference of M. alternatus Adults

The feeding experiments showed that *M. alternatus* adults can feed on young branches (one year old) (Figure 1D), old branches (2–3 years old) (Figure 1E), and bark of the logs (Figure 1F) of the three pine trees for their complementary nutrition. The feeding areas were the largest on the old branches (60.77%), followed by the log bark (22.86%) and the young branches (16.36%) (Appendix A). The total feeding area was the largest on *P. massoniana* (42.43%), followed by those on *P. tabuliformis* (36.94%) and *P. armandii* (20.63%) (Figure 5A, Appendix A). Within a given pine species, the feeding areas on young branches, old branches and logs were also significantly different (Figure 5A). As for *P. tabuliformis*, the feeding area was larger on the old branch bark, accounting for 69.15%, than on the young branch bark (22.44%) and the log bark (8.41%). On *P. massoniana*, the feeding area was larger on the old branch bark (44.07%) than those on the log bark (39.53%) and the young branch bark (16.40%). On *P. armandii*, the feeding area was larger on the old branch bark (80.12%) than those on the log bark (14.49%) and the young branch bark (5.39%) (Appendix A).

### 3.4. Oviposition Selection of M. alternatus Adults

The oviposition experiments showed that the shape of the oviposition scars differed significantly among the host pine species, with different diameters at breast height. They were round on the host pine trees, with a smaller diameter at breast height (Figure 1G), while they were oval on the host pine trees, with a larger diameter at breast height (Figure 1H). A single egg was laid under each oviposition scar, generally 1 cm below the oviposition scar (Figure 1I). The oviposition scars on *P. armandii* accounted for 42.91%, followed by those on *P. tabuliformis* (33.74%) and *P. massoniana* (23.36%) (Figure 5B, Appendix A).

## 4. Discussion

In the present study, we found that the population density of *M. alternatus* was significantly higher on *P. armandii* than on *P. massoniana* and *P. tabuliformis*, and the developmental stages of *M. alternatus* larvae were the largest on *P. armandii*, followed by those on *P. massoniana* and *P. tabuliformis*. It was revealed that many individuals of *M. saltuarius* that hatched in June and July became fourth-instar larvae in November, while the populations that hatched in August developed into third-instar larvae (60–70%) or fourth-instar larvae (30–40%) in November, and the populations that hatched in September developed into first- to third-instar larvae in November [24]. *M. saltuarius* larvae showed different growth and development on the conifers of *P. koraiensis*, *P. densiflora*, *Abies holophylla*, *Larix leptolepsis*, *P. bungeana*, and *P. rigida*, respectively [24]. In addition, the size of *M. saltuarius* larvae was the largest on *P. bungeana*, mid-sized on *P. koraiensis*, *P. densiflora* and *A. holophylla*, and the smallest on *L. leptolepsis* and *P. rigida* [25]. It has been suggested that pre–diapause larvae of *M. alternatus* undergo intense diapause in the early summer, while the remaining individuals undergo weak diapause in the early autumn [26]. The voltinism and the ovariole number of females of *M. alternatus* were not affected by the pine tree species [27]. It was revealed that *M. saltuarius* preferred to oviposit on *P. densiflora,* followed by *P. koraiensis, P. rigida,* and *Larix leptolepis* [28]. This could be due to the bark thickness of different host pine species. The mass indoor breeding of *M. alternatus* adults on *P. massoniana* was beneficial to the reproduction of offspring when compared with those feeding on *P. elliottii* and *Cedarus cedarosa* [13]. In the present study, we revealed that *M. alternatus* preferred to oviposit on *P. armandii*, followed by *P. tabuliformis* and *P. massoniana*, which corresponds to the fact that the development time of *M. alternatus* larvae was significantly earlier on *P. armandii* than on *P. massoniana* and *P. tabuliformis*. Our findings, combined with prior studies, indicate that the oviposition preference of *M. alternatus* for different host pines may affect the population density and developmental stage of *M. alternatus* larvae. The host preference of *M. alternatus* should have corresponding succession with the pine species succession in different geographical environments, which may have a certain order due to the great diversity of host pine species in the north and south of China.

Controversially, it has been reported that *M. alternatus* has four instars or five instars [29,30,31,32,33]. Dyar’s law could help us to infer the actual molting times of a given insect from discontinuous or incomplete molting material [34,35]. However, we revealed that the instars of *M. alternatus* larvae cannot be accurately determined, because the head capsule width and the pronotum width of *M. alternatus* larvae increase continuously during their ontogenetic development. This is consistent with the results of a previous study [33].

Our results show that among the three pine tree species, *P. massoniana* was the most popular food source and *P. armandii* the most unpopular food source of *M. alternatus* adults for the purpose of complementary nutrition. However, we revealed that *M. alternatus* preferred to oviposit on *P. armandii*. We also found that the shape of the oviposition scars was round on host pine trees with a smaller diameter at breast height but oval on hosts with a larger diameter at breast height. This could also be related to the bark thickness and the growth conditions of the host trees. *M. alternatus* adults prefer to feed on healthy pine trees, but they prefer to oviposit on weak wood [36]. The emissions of pine trees include various terpenes, and the compounds are likely to be a particularly important factor in determining the preference of *M. alternatus* [37,38]. In addition, the emission of volatile compounds varies between weak pine trees infested with the pine wood nematode and healthy pine trees [39]. Trees infected by the pine wood nematode emit more terpenes, particularly α-pinene, which is the most attractive compound for *M. alternatus* females [40]. *M. alternatus* adults can be attracted to volatile mixtures of monoterpenes and oxygenated monoterpenes, and the oviposition can be affected by semiochemicals such as volatile plant compounds and insect pheromones [40,41]. Further investigations are required to determine the chemical constituents of weak trees infested with the pine wood nematode and healthy trees.

## 5. Conclusions

The population density of *M. alternatus* was significantly higher on *P. armandii* than *P. massoniana* and *P. tabuliformis* due to the oviposition preference of *M. alternatus* adults. The development of *M. alternatus* larvae was continuous, and the instar of *M. alternatus* could not be accurately determined, because Dyar’s law is not suitable for continuously developing individuals. Our results provide useful information for the comprehensive prevention and control of the pine wilt disease in the Qinling–Daba Mountains and their adjacent areas.

## Figures and Tables

**Figure 1 insects-14-00181-f001:**
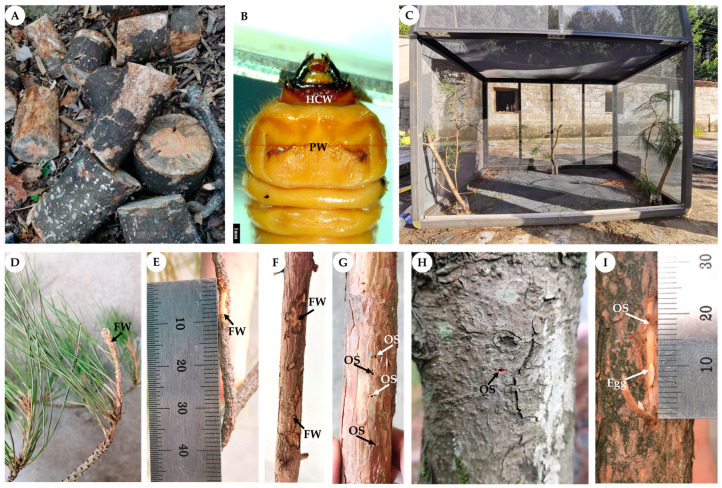
Collection and investigation of *Monochamus alternatus* on different host plants. (**A**): Logs of *P. armandii* damaged by the pine wood nematode and *M. alternatus*. (**B**): Measurement of *M. alternatus* larvae (HCW: head capsule width; PW: pronotum width). (**C**): Net cages containing branches and logs of *P. tabuliformis*, *P. armandii*, and *P. massoniana*. (**D**): Feeding wound (FW) on the young branches (one year old) caused by *M. alternatus*. (**E**): Feeding wound on the old branches (2–3 years old) caused by *M. alternatus*. (**F**): Feeding wound on the log bark caused by *M. alternatus*. (**G**): Round oviposition scars (OS) caused by *M. alternatus* adults. (**H**): Oval oviposition scars caused by *M. alternatus* adults. (**I**): Distance between eggs and oviposition scars of *M. alternatus*.

**Figure 2 insects-14-00181-f002:**
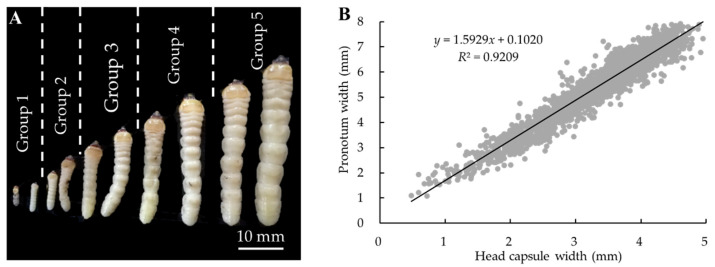
Continuous development of *Monochamus alternatus* larvae. (**A**): A continuous increase in the body size of *M. alternatus* larvae. (**B**): A continuous increase in the head capsule width and the pronotum width of *M. alternatus* larvae.

**Figure 3 insects-14-00181-f003:**
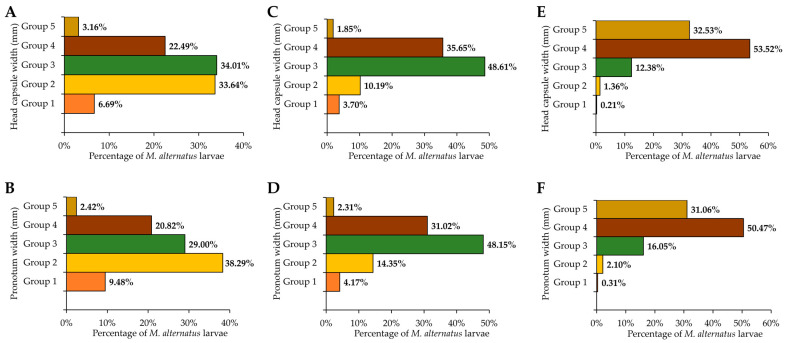
Population density of *Monochamus alternatus* larvae of different development stages on three pine trees. (**A**,**B**): *Pinus tabuliformis*. (**C**,**D**): *Pinus massoniana*. (**E**,**F**): *Pinus armandii*.

**Figure 4 insects-14-00181-f004:**
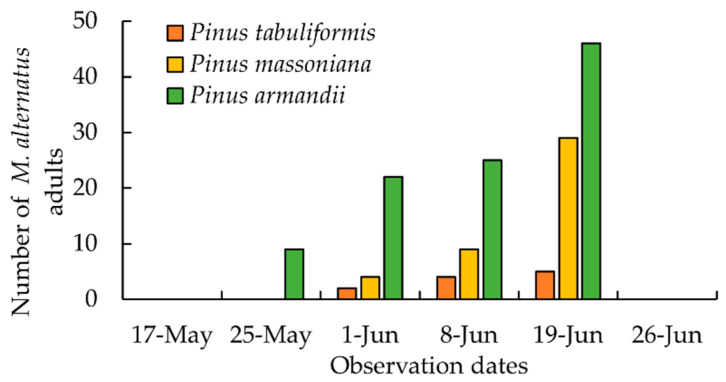
The number of *Monochamus alternatus* adults in net cages on different observation dates.

**Figure 5 insects-14-00181-f005:**
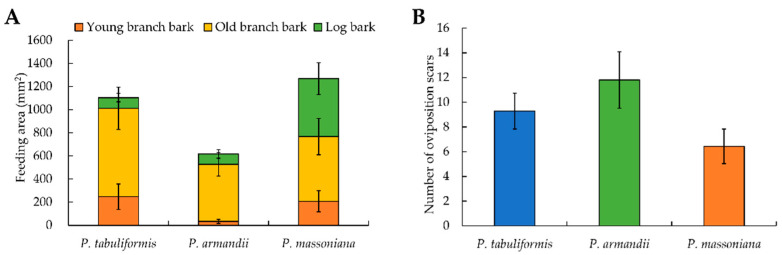
Feeding and oviposition preference of *Monochamus alternatus* adults on three pine trees. (**A**): The feeding area of *M. alternatus* on different parts of three kinds of pine trees. (**B**): The number of oviposition scars of *M. alternatus* adults on three pine trees.

**Table 1 insects-14-00181-t001:** Surveying sites of *Monochamus alternatus* in the Qinling–Daba Mountains of China.

Surveying Sites			Tree Species	Longitude (E)	Latitude (N)	Altitude (m)
Qinling Mountains	Zhashui County	Fenghuang Town	*P. tabuliformis*, *P. armandii*	109°21′34″	33°43′59″	659.00
		Yingpan Town	*P. armandii*	109°06′03″	33°74′21″	931.33
		Xialiang Town	*P. tabuliformis*	109°08′54″	33°37′42″	736.06
	Foping County	Daheba Town	*P. tabuliformis*, *P. massoniana*	108°61′34″	33°29′43″	769.05
	Ningshan County	Chengguan Town	*P. tabuliformis*, *P. armandii*	108°19′48″	33°18′00″	807.02
Daba Mountains	Xixiang County	Yankou Town	*P. massoniana*	107°50′42″	32°59′46″	509.88
	Langao County	Chengguan Town	*P. massoniana*	108°54′35″	32°18′30″	758.63
		Zuolong Town	*P. massoniana*	108°53′08″	32°30′19″	457.31

**Table 2 insects-14-00181-t002:** Population density of *Monochamus alternatus* larvae.

Tree Species	N	Number of *Monochamus alternatus* Larvae ± SE	Variance Analysis
*Pinus tabuliformis*	10	73.00 ± 21.81 b	*F* = 10.480, *p* = 0.000
*Pinus massoniana*	14	91.21 ± 18.80 b	
*Pinus armandii*	14	235.50 ± 36.06 a	

Notes: The different letters indicate a significant difference (*p* < 0.01).

## Data Availability

Data are contained within the article and Appendix A.

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
