# Peer review of "Population Density and Host Preference of the Japanese Pine Sawyer (Monochamus alternatus) in the Qinling–Daba Mountains of China"

_insects, 2023, doi:10.3390/insects14020181_

Round 1
Reviewer 1 Report
The present manuscript deals into the population density of M. alternatus overwintering larvae and host preference of M. alternatus adults on different host plants (Pinus tabuliformis, P. armandii, and P. massoniana) killed by pine wilt disease.
Although is an interesting study, I recommend some improvements or clarifications especially in the results.
Here in the pdf I want highlight also these modifications/suggestions should be made prior to its publication.
Most important is include the sequences information (COI) or exclude from the paper and the Methodology. A figure-map would accomplish the visual understanding, the use acronym I believe is unnecessary and within the discussion the different cites researches may be integrated in the text with your results avoiding and Introduction structure.
Thanks in advance for answer my questions and doubts and congratulations for the realization of this interesting research.
Best regards,
P. M-P

Author Response
Response to Reviewer 1 Comments
Point 1: Here in the pdf I want highlight also these modifications/suggestions should be made prior to its publication.
Response: Thank you very much for your suggestions. We have revised related text in the manuscript.
Point 2: Most important is include the sequences information (COI) or exclude from the paper and the Methodology.
Response: Thank you very much for your suggestion. we have revised this section (Lines 108-113).
Point 3: A figure-map would accomplish the visual understanding, the use acronym I believe is unnecessary and within the discussion the different cites researches may be integrated in the text with your results avoiding and Introduction structure.
Response: Thank you very much for your suggestion. We did not provide a figure-map but re-organized Table 1 and revised related text in “2.1 Experimental field”. We have replaced all the acronyms, and different cities mentioned in the discussion were integrated in the text with the Introduction and Results.
Reviewer 2 Report
The paper “Population density and host preference of the Japanese pine sawyer (Monochamus alternatus) in the Qinling-Daba Mountains of China“ has scientific potential due to the new data on the ecology and development of longhorn beetle, causing serious damage to forest ecosystems in the European Union, as well as in Asian countries.
However, I propose to supplement the text with some information, some data could be improved or/and well thought-out.
Introduction:
- In the first paragraph the Authors write about the pine wilt disease (PWD) caused by the pine wood nematode (PWN), Bursaphelenchus xylophilus. The PWN is not the hero of that paper so is a little confusing that the beginning of the paper deals with nematode, not insect.
- The fragment “It was necessary to better understand the influence of different hosts on the dispersal behavior of this species [12]. The effective control of M. alternatus can effectively limit the spread of PWD in QDM.” is unclear.
- “The purpose of this study was to determine the damage degree of M. alternatus to different pine hosts and the host selection”. Please clarify and elaborate how the damage degree was determined, so is unclear.
- The authors are asked to provide a criterion for the selection of pine species: P. tabuliformis, P. armandii, and P. massoniana.
Materials and methods
- Table 1. Surveying sites of Monochamus alternatus in the Qinling-Daba Mountains of China.
In addition to these data, it would be beneficial to provide precise data on the distribution of individual specimens of trees under study, i.e. the distance from each other of the individual specimens of the tree species under study. This is important with regard to the type of distribution of insects, which often occur clustered. A study of the abundance of insects populations inhabiting trees that are in close proximity and farther away may yield different results.
- 2.2. Identification of M. alternatus on different host plants
The methods of identification should be divided on imago and larvae and described for two developmental stages separately. It could be useful to give the name of other species of Cerambycidae with which M. alternatus (imago and larvae) may be confused and provide literature for identification.
- 2.3. Population density and development stages of M. alternatus on different host plants
“According to the distribution of dead pine trees caused by PWD in the study areas, at least 10 trees… of each pine trees infected by PWN and died in the same year were selected in each study site in December of 2020”
The number of trees under study “at least 10“ is very confusing. Whether the number of trees should not be strictly standardised for each of the three species. How did the Authors relate the number of larvae captured from each pine species to the number of tree individuals? The number of tree of each species should be given here (not only in results).
Please give the criteria for the choice of research time.
- “Then, we stripped the bark of the logs, and split the logs into small splints with the diameter of no more than 2 cm…”
Very controversial aspect of the methodology “no more than 2 cm”. Please be convinced that the pieces of wood were not too large in relation to the size of the smallest larvae (youngest larval stages). It can be surmised that the methodology chosen in this way resulted in the youngest individuals being overlooked and missed.
- Figure 1.
Lack of scale bar on some photos. Each photo should be mentioned/ briefly described in the text.
Results
- Figure 2. The group of larval stages should be marked on the photos.
- 3.2. The emergence of M. alternatus adults on different pine tree.
The is no information about this part of the study in the methodology. Please there is no information about this part of the research in the methodology. Please complete the methodology.
Discussion
- “Here, we found that the population density of M. alternatus was significantly higher on P. armandii than P. massoniana and P. tabuliformis …”
Such a conclusion can be made provided that the pieces of wood examined were small enough to condition the accurate selection of all individuals especially of the smallest larvae (Group 1-2). As can be seen from the data on the size of the collected larvae: "Group 1: HCW 0.51 ~ 1.40 mm, PW 1.11 ~ 2.50 mm; 184; Group 2: HCW 1.41 ~ 2.30 mm, PW 2.51 ~ 3.90 mm" their head width, pronotum width and body lengths indicate that they may have been missed during the collection of individuals and affect significantly the reliability of the data obtained"
- “… and developmental stages of M. alternatus larvae were the largest on P. armandii, followed by P. massoniana and P. tabuliformis”.
Could the authors comment on what might account for this relationship?
Author Response
Response to Reviewer 2 Comments
Point 1: In the first paragraph the Authors write about the pine wilt disease (PWD) caused by the pine wood nematode (PWN), Bursaphelenchus xylophilus. The PWN is not the hero of that paper so is a little confusing that the beginning of the paper deals with nematode, not insect.
Response: Thank you very much for your comments and suggestion. We have revised the Introduction thoroughly.
Point 2: The fragment “It was necessary to better understand the influence of different hosts on the dispersal behavior of this species [12]. The effective control of M. alternatus can effectively limit the spread of PWD in QDM.” is unclear.
Response: Thank you very much for your comment. We have revised this sentence in the updated manuscript (Lines 73–74).
Point 3: “The purpose of this study was to determine the damage degree of M. alternatus to different pine hosts and the host selection”. Please clarify and elaborate how the damage degree was determined, so is unclear.
Response: Thank you very much for your comment. We are ashamed for this mistake and have deleted this sentence, because the determination of damage degree had never been conducted.
Point 4: The authors are asked to provide a criterion for the selection of pine species: P. tabuliformis, P. armandii, and P. massoniana.
Response: Thank you very much for your comment. The criterion for the selection of these three pine species is due to that they are the main susceptible pine species in the Qinling–Daba Mountains, which has been described in Lines 62–65.
Point 5: Table 1. Surveying sites of Monochamus alternatus in the Qinling-Daba Mountains of China.
In addition to these data, it would be beneficial to provide precise data on the distribution of individual specimens of trees under study, i.e. the distance from each other of the individual specimens of the tree species under study. This is important with regard to the type of distribution of insects, which often occur clustered. A study of the abundance of insects populations inhabiting trees that are in close proximity and farther away may yield different results.
Response: Thank you very much for your suggestion. We have added have added related information in lines 103-105.
Point 6: 2.2. Identification of M. alternatus on different host plants. The methods of identification should be divided on imago and larvae and described for two developmental stages separately. It could be useful to give the name of other species of Cerambycidae with which M. alternatus (imago and larvae) may be confused and provide literature for identification.
Response: Thank you very much for your suggestion. We have revised this section (Lines 108-113).
Point 7: 2.3. Population density and development stages of M. alternatus on different host plants.“According to the distribution of dead pine trees caused by PWD in the study areas, at least 10 trees… of each pine trees infected by PWN and died in the same year were selected in each study site in December of 2020”. The number of trees under study “at least 10“ is very confusing. Whether the number of trees should not be strictly standardised for each of the three species. How did the Authors relate the number of larvae captured from each pine species to the number of tree individuals? The number of tree of each species should be given here (not only in results).
Response : Thank you very much for your suggestion. The number of tree of each species have added (Line 116–117).
Point 8: Please give the criteria for the choice of research time.
Response: Thank you very much for your suggestion. The word “selected” was changed to “sampled”. We conducted our field sampling in December because the dead trees caused by the pine wilt disease in this arera could be easily identified.
Point 9: Then, we stripped the bark of the logs, and split the logs into small splints with the diameter of no more than 2 cm…” Very controversial aspect of the methodology “no more than 2 cm”. Please be convinced that the pieces of wood were not too large in relation to the size of the smallest larvae (youngest larval stages). It can be surmised that the methodology chosen resulted in the youngest individuals being overlooked and missed.
Response: Thank you very much for your suggestions. We have revised rtelated text in the methodology (lines 121-124).
Point 10: Figure 1. Lack of scale bar on some photos. Each photo should be mentioned/briefly described in the text.
Response: Thank you very much for your suggestions. We have updaed the images according the suggestions of other reviewers, and a scale bar is provided in all the photos.
Point 11: Figure 2. The group of larval stages should be marked on the photos.
Response: Thank you very much for your suggestion. The group of larval stages have been marked on the photos.
Point 12: 3.2. The emergence of M. alternatus adults on different pine tree. The is no information about this part of the study in the methodology. Please there is no information about this part of the research in the methodology. Please complete the methodology.
Response: Thank you very much for your suggestions. Related information have been given in the the methodology (Lines 129–133).
Point 13: Discussion. Here, we found that the population density of M. alternatus was significantly higher on P. armandii than P. massoniana and P. tabuliformis …”. Such a conclusion can be made provided that the pieces of wood examined were small enough to condition the accurate selection of all individuals especially of the smallest larvae (Group 1-2). As can be seen from the data on the size of the collected larvae: "Group 1: HCW 0.51 ~ 1.40 mm, PW 1.11 ~ 2.50 mm; 184; Group 2: HCW 1.41 ~ 2.30 mm, PW 2.51 ~ 3.90 mm" their head diameters, forelimb diameters and body lengths indicate that they may have been missed during the collection of individuals and affect significantly the reliability of the data obtained.
Response: Thank you very much for your suggestions. Related text have been added in the methodology (Lines 121–124).
Point 14: … and developmental stages of M. alternatus larvae were the largest on P. armandii, followed by P. massoniana and P. tabuliformis”. Could the authors comment on what might account for this relationship?
Response: Thank you very much for your suggestions. The factors that might account for this relationship have been discussed in the updated version.
Reviewer 3 Report
The manuscript by Nan et al. present a study of the population density of Monochamus alternatus on three pine speciese: Pinus tabuliformis, P. armandii, and P. massoniana with a study conducted in the Qinling-Daba Mountains (China). Moreover, also feeding preferences on these three plant species were analized. M. alternatus is one of the main vectors of the nematode causing the Pine wild disease (PWD) and more accurate knowledge is needed to counter the spread of this disease.
The manuscript and the results presented are very interesting. However, there are some points that, in my opinion, need to be improved or implemented. The main issue I have with the manuscript regards statistical analyses: it seems that authors conduct the analysis of variance (ANOVA) only on data concerning the population density (Paragraph 3.1), while statistics are missing for data on, for example, the feeding preference (Paragraph 3.3) or the oviposition selection (Paragraph 3.4). Authors must also include in the manuscript the statistical outputs of these other sections.
I also have some minor comments and corrections.
Line 62: correct “tree species are P. massoniana.” With “tree species is P. massoniana.”
Table 1 seems really difficult for me to understand (probably because of my lack of knowledge of the physical and political geography of China). In particular, the first three columns regarding surveying sites: how are they organized? What are the three bold locations in the first row? Why are some cells in the other rows blank? Could you better explain the organization of the study sites, correcting not only the table but also integrating an explanation in the text (Lines 94-97)?
Paragraph 2.3 describe, again, the method of identification of M. alternatus (albeit in less detail) [lines 115-116]. I suggest to include the paragraph 2.2 in paragraph 2.3 and delete the former.
Images composing Figure 1 are too small and therefore difficult to see. I suggest to reduce the number of images, eliminating unnecessary ones (e.g., 1A, 1C, 1F or 1G, 1H)
Lines 137-144: please correct the first sentence of the paragraph in this way: “…three pairs of newly emerged males and females of M. alternatus adults were placed into a cage (1.5 m × 1.0 m × 1.0 m) in Chengguan Town, Ningshan County and provided with fresh pine branches (length: 92.2 ± 1.6 cm), inserted into bottles filled with water, and logs (length: 71.3 ± 0.7 cm, diameter: 26.3 ± 3.1 cm) of three pine species, aiming…”
Line 146: please correct “and total 23 times were repeated” with “for a total amount of 23 repetitions”
Line 148: please correct “trees” with “species”
In lines 149-157 you talk about “young branches” and “old branches”, but this distinction has never been mentioned before. Could you better explain in lines 137-144 this aspect? For example, young and old branches of the same pine species were placed together in the same bottle, or you set one bottle for each combination of young-old branches and pine species?
Generally, the “Results” section only presents the results, without commenting on them. I suggest limiting comments on the results (e.g., lines 169-170, 190-191, 203-204, 225-226, 228-230, 242-243) to the "Discussion" section.
Line 216: Correct with “…adults can feed on young branches…”
Line 250: Correct with “…larvae and adults feeding on P. bungeana…
Pay attention to the citation form, present in different ways (apex or not).
Lines 290-293: This sentence is not clear: maybe you want say “Kim and Kim (2020) indicated that determining the number of instars in insect species by applying the frequency distribution of HCW can mislead actual instars numbers, since HCW of successive instars largely overlaps in the field”? Please clarify the sentence.
Please double-check the references. First they are doubled, and then some errors such as some authors written in bold should be corrected.
Finally, although not a native speaker, I believe it is best to have the manuscript proofread by a native speaker to improve English and sentence structure, which is often difficult to understand.
In conclusion, the manuscript is generally well-written but, to be published, it needs some minor corrections and refinements, especially in the Materials and Methods.
Author Response
Response to Reviewer 3 Comments
Point 1: The manuscript and the results presented are very interesting. However, there are some points that, in my opinion, need to be improved or implemented. The main issue I have with the manuscript regards statistical analyses: it seems that authors conduct the analysis of variance (ANOVA) only on data concerning the population density (Paragraph 3.1), while statistics are missing for data on, for example, the feeding preference (Paragraph 3.3) or the oviposition selection (Paragraph 3.4). Authors must also include in the manuscript the statistical outputs of these other sections.
Response: Thank you very much for your suggestions. We have added the statistical outputs of the feeding preference (Paragraph 3.3) and the oviposition selection (Paragraph 3.4) in the manuscript as well as Table S3 of Supplementary Files.
Point 2: I also have some minor comments and corrections. Line 62: correct “tree species are P. massoniana.” With “tree species is P. massoniana.”
Response: Thank you very much for your suggestion, and related text haves revised.
Point 3: Table 1 seems really difficult for me to understand (probably because of my lack of knowledge of the physical and political geography of China). In particular, the first three columns regarding surveying sites: how are they organized? What are the three bold locations in the first row? Why are some cells in the other rows blank? Could you better explain the organization of the study sites, correcting not only the table but also integrating an explanation in the text (Lines 94-97)?
Response: Thank you very much for your suggestions. We have re-organized Table 1 and revised related text in “2.1 Experimental field”.
Point 4: Paragraph 2.3 describe, again, the method of identification of M. alternatus (albeit in less detail) [lines 115-116]. I suggest to include the paragraph 2.2 in paragraph 2.3 and delete the former.
Response: Thank you very much for your suggestions. We have revised the section related to the method of identification of M. alternatus.
Point 5: Images composing Figure 1 are too small and therefore difficult to see. I suggest to reduce the number of images, eliminating unnecessary ones (e.g., 1A, 1C, 1F or 1G, 1H)
Response: Thank you very much for your suggestions. We have updated Figure 1.
Point 6: Lines 137-144: please correct the first sentence of the paragraph in this way: “…three pairs of newly emerged males and females of M. alternatus adults were placed into a cage (1.5 m × 1.0 m × 1.0 m) in Chengguan Town, Ningshan County and provided with fresh pine branches (length: 92.2 ± 1.6 cm), inserted into bottles filled with water, and logs (length: 71.3 ± 0.7 cm, diameter: 26.3 ± 3.1 cm) of three pine species, aiming…”
Response: Thank you very much for your suggestions. We have revised this sentence.
Point 7: Line 146: please correct “and total 23 times were repeated” with “for a total amount of 23 repetitions”; Line 148: please correct “trees” with “species”.
Response: Thank you very much for your suggestions. We have revised these sentences.
Point 8: In lines 149-157 you talk about “young branches” and “old branches”, but this distinction has never been mentioned before. Could you better explain in lines 137-144 this aspect? For example, young and old branches of the same pine species were placed together in the same bottle, or you set one bottle for each combination of young-old branches and pine species?
Response: Thank you very much for your suggestions. We have revised related texts in the updated manuscript (lines 175-180).
Point 9: Generally, the “Results” section only presents the results, without commenting on them. I suggest limiting comments on the results (e.g., lines 169-170, 190-191, 203-204, 225-226, 228-230, 242-243) to the "Discussion" section.
Response: Thank you very much for your suggestions. We have revised related texts in the Results section.
Point 10: Line 216: Correct with “…adults can feed on young branches…”; Line 250: Correct with “…larvae and adults feeding on P. bungeana…; Pay attention to the citation form, present in different ways (apex or not).
Response: Thank you very much for your suggestions. We have revised these sentences.
Point 11: Lines 290-293: This sentence is not clear: maybe you want say “Kim and Kim (2020) indicated that determining the number of instars in insect species by applying the frequency distribution of HCW can mislead actual instars numbers, since HCW of successive instars largely overlaps in the field”? Please clarify the sentence.
Response: Thank you very much for your suggestions. We have revised this sentence.
Point 12: Please double-check the references. First they are doubled, and then some errors such as some authors written in bold should be corrected.
Response: Thank you very much for your suggestions. The references have been double-checked, and mistakes have been corrected.
Round 2
Reviewer 2 Report
Thank you to the Authors for responding to the comments.
All the best